# Neglected Tropical Diseases

# Hiding in plain sight: Genomic and phenotypic characterization of mosquito-borne Bussuquara virus

Madeline R. Steck[1], Cecília A. Banho[2], Vsevolod L. Popov[1], Haiping Hao[3], Kathryn A. Hanley[4], Mauricio L. Nogueira[1,2], Nikos Vasilakis[1,5*]

1 Department of Pathology, University of Texas-Medical Branch, Galveston, Texas, United States of America, 2 Laboratório de Pesquisas em Virologia, Faculdade de Medicina de São José do Rio Preto, São José do Rio Preto, São Paulo, Brazil, 3 Department of Biochemistry & Molecular Biology, University of Texas-Medical Branch, Galveston, Texas, United States of America, 4 Department of Biology, New Mexico State University, Las Cruces, New Mexico, United States of America, 5 Center for Vector-Borne and Zoonotic Diseases, University of Texas Medical Branch, Galveston, Texas, United States of America

* nivasila@utmb.edu

## Abstract

Bussuquara virus (BSQV), an orthoflavivirus discovered in Brazil in 1956, has been detected throughout the Americas in diverse mosquito and vertebrate species, including humans. Critical gaps in BSQV knowledge include its capacity for urban transmission and clinical pathogenesis outcomes, with insufficient historical experimentation to draw genomic or phenotypic comparisons to related orthoflavivirus species. The objective of this study was to conduct morphologic, genomic, phylogenetic, and *in vitro* viral fitness characterization of BSQV using the four available historical strains. We used next generation sequencing and rapid amplification of cDNA ends to construct consensus genomes, followed by phylogenetic analysis and genome annotation to evaluate orthoflavivirus evolutionary relationships and genome characteristics. Infected mosquito (C6/36) and non-human primate (Vero CCL81) cells were imaged with transmission electron microscopy. Viral replication kinetics were quantified across seventeen cell lines of mosquito, mammal, rodent, avian, non-human primate, and human origin. BSQV morphologic (virion diameter, cytopathic effect) and genomic (size, organization, architecture, sequence motifs) results were in line with canonical orthoflavivirus characteristics. One of the four strains (CoAr 41922) shared greater sequence homology to the Naranjal orthoflavivirus than other BSQV strains and was thus excluded from infection phenotype experiments. All three confirmed BSQV strains replicated robustly in most mosquito and all vertebrate cell lines, causing either minimal (mosquito) or moderate to extreme (vertebrate) cytopathic effects. We conclude that BSQV is a generalist orthoflavivirus with a broad range of susceptible vertebrate and mosquito vectors. Our data build a foundation for pathogenesis

**Data availability statement:** All raw data described in this paper (S1 File, S1 and S4 Tables) and the results of the statistical analysis (S5 Table) and genome annotation (S2 and S3 Tables) are available for access through Zenodo (doi:10.5281/zenodo.15794191).

**Funding:** This research was supported by the Centers for Research in Emerging Infectious Diseases (CREID), "The Coordinating Research on Emerging Arboviral Threats Encompassing the Neotropics (CREATE-NEO)" grant U01AI151807, awarded to NV/KAH by the National Institutes of Health. MRS acknowledges support by a T32 training grant (T32 AI007526-24) on Emerging and Tropical Infectious Diseases Training Program by the National Institutes of Health. This work was supported in part from the Fundação de Amparo à Pesquisa do Estado de São Paulo (FAPESP, grant numbers 2022/03645-1 to MLN, 2023/14670-0 to CAB). MLN is also supported by INCT Viral Genomic Surveillance and One Health by CNPq grant 4057586/2022-0. M.L.N. is a CNPq Research Fellow. The funders had no role in the study design, data collection and analysis, decision to publish, or preparation of the manuscript.

and vector competence studies to determine the potential of BSQV to emerge into epizootic and urban transmission cycles.

## Author summary

After decades of being considered negligible pathogens, multiple mosquito-borne viruses, such as Zika and Mayaro, are now recognized as significant human health threats across the neotropics. Bussuquara virus (BSQV) is a mosquito-borne orthoflavivirus discovered in Brazil that is closely related to epidemic orthoflavivirus species; however, the current and potential future threats posed by BSQV to human health are unknown. Our study is the first assessment into BSQV's capacity to become a medically important mosquito-borne virus in the Americas, and we do so by characterizing the BSQV virion morphology, genome, and evolutionary relationships and comparing them to their more well-studied orthoflavivirus relatives. We also investigate the ability of BSQV to infect or impair representative mosquito and vertebrate cells. Our findings demarcate the genomic similarities and differences, implications yet unknown, between BSQV and more prominent orthoflaviviruses. Cell infection studies suggest that BSQV can infect a broad range of mosquito and vertebrate species, with cytopathy uncommon in mosquito cells but prominent in vertebrate cells, including those of human brain, testes, and kidney origins. These data suggest BSQV has the capacity to sustain transmission cycles within human populations by expansion of its vector host range, and to cause human disease.

## Introduction

Bussuquara virus (Family: *Flaviviridae*, Genus: *Orthoflavivirus*) was discovered in 1956 in the Amazon rainforest in northern Brazil [1]. BSQV isolations and antibody detection were then reported from multiple mosquito genera (*Culex, Trichoprosopan, Mansonia, Collquilettidia*) and vertebrate taxa (non-human primates, rodents, birds, sloths, bats, ruminants, horses, and humans) [2]. To date, detections are limited to sylvatic and epizootic transmission cycles and there is no reported evidence of urban transmission between domesticated mosquitoes (i.e., *Aedes aegypti* and *Ae. albopictus*) and humans. However, BSQV is known to infect and cause disease in humans, as demonstrated by a self-resolving febrile case in 1970s Panama [3] and detection of convalescent IgG immunity in humans from rainforest-proximate villages in Panama [3] and Argentina [4]. The spectrum of potential clinical manifestations is undefined, although an early animal study [5] demonstrated fatal central nervous system virulence in neonatal hamsters.

The neotropics are hyperendemic for many arthropod-borne viruses (arboviruses); however, only those designated as medically relevant are intentionally surveilled or diagnosed in clinical settings. Multiple mosquito-borne orthoflaviviruses have emerged in the Americas including Zika (ZIKV), dengue (DENV) serotypes 1–4, Ilhéus (ILHV), yellow fever (YFV), Saint Louis encephalitis (SLEV), and Rocio (ROCV) viruses [6–12]. BSQV

is not currently a priority for surveillance, and its detection is further hindered by a lack of commercial diagnostics. Thus, the true infection burden of BSQV in human populations is unknown, since low-level circulation could be masked by inapparent or undifferentiated arboviral symptoms or orthoflavivirus cross-reactivity in serological assays [13].

It is critical to proactively assess the emergence risk of neglected arboviruses. Recent decades illustrate the challenges of mitigating unforeseen arbovirus emergence events including accurate diagnosis, interrupting transmission, or providing effective therapeutics [14–16]. With five hundred arbovirus species listed in the US Centers for Disease Control Arbovirus Catalog (ArboCat) [17], comprehensive characterization of each virus is not feasible. A rational approach is to prioritize arbovirus species with potential to (i) sustain urban transmission and (ii) threaten human health. The historical literature implicates BSQV as a generalist *Culex*-vectored orthoflavivirus with a broad range of susceptible vertebrate species, including humans [2]. Here we sought to evaluate this interpretation of early BSQV studies by analyzing its genome, infection phenotype, evolutionary relationships, and infection kinetics using all available BSQV isolates.

## Materials and methods

### Cells and viruses

A panel of mosquito (n = 7) and vertebrate (n = 10) cell lines were used to assess BSQV replication kinetics (Table 1). Vertebrate and mosquito cells were maintained using the maintenance media described below in a humidified incubator with 5% $CO_2$, at 37°C or 28°C, respectively, excluding CxTr and CT cells which were maintained at 28°C without $CO_2$ or humidity. Infection media used 2% FBS, except HSerC cell infection media which retained 10% FBS. CxTr was obtained from the Foreign Arthropod-Borne Animal Disease Research Unit (US Department of Agriculture, Manhattan, KS, USA), while Hsu and CT were both gifted by the Ebel lab at Colorado State University, Fort Collins, CO. HserC cells were purchased from iXCells Biotechnologies (San Diego, California, USA). All other cell lines originated from the American Type Culture Collection (ATCC, Manassas, VA, USA).

Three BSQV strains (BeAn 217201, BeAn 4116, CoAr 41922) were obtained from the World Reference Center for Emerging Viruses and Arboviruses (University of Texas Medical Branch, Galveston, TX, USA). The prototype BSQV strain, BeAn 4073, was sourced from ATCC. Passage history of the lyophilized stocks upon receipt is noted in Table 2. Vero CCL81 cells were infected with each strain for one passage until >50% ablation of the monolayer (2–3 days). Infected cell culture supernatant was collected, clarified through centrifugation, supplemented with 1X SPG buffer, and stored at -80°C until experimental use. Two strains (BeAn 4116, BeAn 217201) had one extra passage in Vero CCL81 cells supplemented with 25µg/ml Plasmocin antibiotic (InvivoGen, San Diego, CA, USA) before stock expansion.

### Virus quantification

Vero CCL81 cells were seeded onto 24-well plates to reach ~80% confluency by the following day (~24 hours). Monolayers were then infected with 100µL/well of inoculum, with triplicate wells of sample dilutions ($10^{00}$-$10^5$) in infection media (**Table 1**) per virus strain and day. Plates were incubated at 37°C with 5% $CO_2$ for one hour with rocking every fifteen minutes. A methylcellulose overlay - 0.8% methyl cellulose (Sigma-Aldrich), 1X MEM, 3% FBS, 1% P/S - was added to cells for 3 (BeAn 217201) or 4 (BeAn 4116, BeAn 4073) day incubation. The methylcellulose overlay was then removed, and monolayers were fixed with 1:1 acetone:methanol solution at room temperature for one hour, followed by washing and drying. All wells were stained with 0.2% (vol/vol) crystal violet (Sigma-Aldrich) in methanol (Sigma-Aldrich). Plaques were counted, averaged between the three replicates, and quantified as plaque forming units per milliliter supernatant (PFU/mL).

### Viral RNA extraction

Viral RNA was extracted and purified following a standard protocol [19] using Trizol reagent (InvivoGen) and chloroform (Sigma-Aldrich). The final RNA pellet was resuspended (~2ng/µL) in 20µL of RNase/DNase and protease-free water (Ambion, Austin, TX, USA) and frozen at -80°C until sequencing or RACE analyses.

**Table 1. Cell lines and culture media compositions.**

| Cell Line | Species | Common Name | Tissue | Base Media | Supplements |
|---|---|---|---|---|---|
| Vero CCL81 | *Cercopithecus aethiops* | African green monkey | kidney epithelium | DMEM | 10% FBS, 1% P/S |
| FRhL-2 | *Macaca mulatta* | Rhesus macaque | lung fibroblasts | EMM | 10% FBS, 1% P/S |
| BHK-21 | *Mesocricetus auratus* | Golden Syrian hamster | kidney fibroblasts | DMEM | 10% FBS, 1% P/S |
| OK | *Didelphis marsupialis virginiana* | North American opposum | kidney epithelium | MEM | 10% FBS, 1% P/S |
| Huh7 | *Homo sapiens* | Human | hepatoma | DMEM | 10% FBS, 1% P/S, 1% NaPy, 1% NEAA |
| SH-SY5Y | *Homo sapiens* | Human | neuroblastoma | DMEM | 10% FBS, 1% P/S |
| HSerC | *Homo sapiens* | Human | human testis | SCBGM | 10% FBS, 1% AA |
| A549 | *Homo sapiens* | Human | lung epithelium | DMEM | 10% FBS, 1% P/S |
| QM7 | *Coturnix coturnix japonica* | Japanese quail | muscle | M199 | 10% FBS, 1% P/S, 10% TPB |
| LMH | *Gallus gallus* | Chicken | liver epithelium | WMB | 10% FBS, 1% P/S* |
| C6/36 | *Ae. albopictus* | Mosquito | larvae | DMEM | 10% FBS, 10% TPB,1% P/S, 1% AmpB |
| C7-10 | *Ae. albopictus* | Asian tiger mosquito | larvae | DMEM | 10% FBS, 1% P/S, 1% NaPy, 1% NEAA |
| U4.4 | *Ae. albopictus* | Asian tiger mosquito | larvae | M&M | 20% FBS, 5% TPB, 1% P/S, 0.15% NaHCO$_3$. |
| Aag2 | *Ae. aegypti* | Yellow fever mosquito | larvae | DMEM | 20% FBS, 2% NaHCO$_3$, 1% TPB, 1% P/S, 1% NEAA |
| Hsu | *Cx. quinque-fasciatus* | Southern house mosquito | ovaries | DMEM | 10% FBS, 10% TPB,1% P/S, 1% AmpB |
| CT | *Cx. tarsalis* | Western encephalitis mosquito | embryonated eggs | SDM | 10% FBS, 1% P/S,1% AmpB |
| CxTr | *Cx. tarsalis* | Western encephalitis mosquito | embryonated eggs | SDM | As described in [18] |

Species and tissue origins are noted for each cell line, alongside base media and supplements composing maintenance media. Abbreviations: DMEM = Dulbecco's Modified Eagle's Medium (ThermoFisher Scientific, Grand Island, NY, USA); EMEM = Eagles Minimal Essential Medium with L-glutamine (ATCC); MEM = Minimum Essential Media (ThermoFisher Scientific); SCBGM = Sertoli Cell Basal Growth Medium (iXCells Biotechnologies); SDM = Schneider's Drosophila Media (ThermoFisher Scientific); M&M = Mitsuhashi and Maramorosch Insect Medium (HIMEDIA, Kelton, PA, USA); M199 = Medium 199 with Earl's BSS (ThermoFisher Scientific); WMB = Waymouth's MB 752/1 (ThermoFisher Scientific); FBS = heat inactivated fetal bovine serum (ThermoFisher Scientific); P/S = Penicillin–streptomycin (100 U/mL and 100 µg/mL, respectively)(ThermoFisher Scientific); NaPy = 100mM sodium pyruvate (ThermoFisher Scientific); NEAA = 100x non-essential amino acids)(Sigma-Aldrich, Burlington, MA, USA); TBP = Tryptose phosphate broth (ThermoFisher Scientific); AmpB = 250µg/mL Amphotericin B (ThermoFisher Scientific); NaHCO$_3$ = 7.5% w/v sodium bicarbonate (Corning, Corning, NY, USA); AA = Antibiotic-Antimyocitic (iXCells Biotechnologies).

**Table 2. BSQV strains.**

| Virus strain | Year | Location | Species | Common Name | Passage History |
|---|---|---|---|---|---|
| BeAn 217201 | 1972 | Brazil | *Proechimys guyannensis* | Guyenne spiny rat | SM-1, Vero-2 |
| BeAn 4116 | 1956 | Brazil | *Alouatta belzebul* | Red-handed howler | SM-1, Vero-2 |
| BeAn 4073 | 1956 | Brazil | *Alouatta belzebul* | Red-handed howler | SM-7 |
| CoAr 41922 | 1960 | Colombia | *Culex spp.* | Common House Mosquito | SM-1, Vero-2 |

Isolation year, country, and species origin recorded for each BSQV strain, and passage history upon receipt. Virus stock preparation required one (BeAn 4073) or two (BeAn 217201, BeAn 4116) additional Vero CCL81 passages before experimental use.

## Next generation sequencing

Viral RNA (~0.9µg) was fragmented by incubation at 94°C for 15 minutes in 11.5µL of First Strand Reaction Buffer and Random Primer Mix (New England Biolabs, Ipswich, MA). A sequencing library was prepared from the sample RNA using NEBNext Ultra II RNA Library Preparation Kit for Illumina (New England Biolabs) following the manufacturer's protocol. Samples were sequenced on a MiniSeq (Illumina, San Diego, CA) in paired-end 75 base format. Reads in fastq format were quality-filtered, and any adapter sequences were removed, using Trimmomatic software [20]. The virus genomes were assembled using ABySS version 2.3.4 with a range of different kmer sizes and different number of reads. The assembled contigs were then clustered using CD-HIT version 4.8.1 [21] and the resulting contig sequences were blasted against a custom database (entirety of viral protein sequences in GenBank UniProt database) using blastp and against nr nucleotide data base using blastn of NCBI Blast version 2.11.0. The blast hits were then manually curated to assemble into complete genome. The full set of trimmed reads were then aligned to the assembled genome using BWA version 0.7.17 to provide accurate alignment for variants calling with lofreq version 2.1.3.1 [22]. The full sets of raw reads were also aligned to the assembled genome using bowtie2 version 2.3.4.1 [23] to enable quantification of percent viral sequences across the reads. A total of 11.04, 8.06, 15.82, and 7.82 million read pairs were generated for the samples containing BSQV strains BeAn 217201, BeAn 4116, BeAn 4073, and CoAr 41922, respectively. Read pairs mapping to the virus in each sample comprised ~1.8 million (BeAn 217201, 16.7%), ~1.7 million (BeAn 4116, 21.29%), ~1.1 million (BeAn 4073, 7.0%), and ~890,000 (CoAr 41922, 11.38%), respectively.

## Rapid amplification of cDNA ends (RACE)

The genomic termini of three BSQV strains BeAn 217201, BeAn 4116, and BeAn 4073 were determined using the kits 5' RACE System for Rapid Amplification of cDNA Ends and 3' RACE System for Rapid Amplification of cDNA Ends (Invitrogen, Vilnius, Lithuania) following manual instructions. Viral RNA from each strain was extracted with the Trizol-chloroform method as outlined above. Consensus BSQV primers were generated using DNASTAR LaserGene primer design tool with default parameters. The *E. coli* RNA Poly(A) Polymerase Kit (New England Biolabs) was used to add a polyadenylated tail to the viral RNA prior to cDNA synthesis for 3' RACE. The 3' termini were PCR amplified first using the primer BSQV_3out (5'- GCCACTGCACTGTACTTCCT-3'), followed by a secondary nested PCR amplification with the primer BSQV_3in (5'-TGATCCCAGGCGAAGGACTA-3') in the forward orientation. The thermocycler (Applied Biosystems, California, USA) conditions for both amplification steps are as follows: 94°C for 3min; 94°C for 1min, 55°C for 1min, 72°C for 70sec (35 cycles); 72°C for 10min. The primer BSQV_5out (5'-TCCAACGTCCATTGCCATGA-3') in the reverse orientation was used for first strand synthesis of the 5' termini, followed by cDNA purification using the QIAquick PCR Purification Kit (Qiagen, Maryland, USA), in lieu of purification with the recommended S.N.A.P. columns. The primer BSQV_5in (5'-TGATCCCAGGCGAAGGACTA-3')
in the reverse orientation was used for PCR amplification using the following thermocycler conditions: 94°C for 3min; 94°C for 1min, 62°C for 1min, 72°C for 70sec (35 cycles); 72°C for 10min. The approximately 1090, 250, and 320 nucleotide fragments from amplification with BSQV_3out, BSQV_3in, and BSQV5_in primers, respectively, were separated by 1% agarose gel electrophoresis and purified with the QIAquick Gel Extraction Kit (Qiagen). For Sanger sequencing, PCR products were labeled using BrilliantDye version 3.1 reaction premix (NimaGen, Nijmegen, Netherlands). Labeled products were resolved and read on a SeqStudio 24 Flex Genetic Analyzer (Applied Biosystems) using a 50 cm capillary array and Pop 7 polymer. Results were analyzed using Sequencing Analysis version 8 from Applied Biosystems.

## Genome annotation

A sequence alignment of BeAn 217201, BeAn 4073, and BeAn 4116 was compared against protein annotations in NCBI GenBank accession NC_009026.2 to designate 3' and 5' untranslated regions (UTR) and the open reading frame (ORF) including three structural and seven nonstructural proteins. The NetPhos 3.1 server was used to predict post-translational

phosphorylation of residues (serine, threonine, tyrosine), with scores above 0.8 reported here. The NetNglyc server was used to predict post-translational glycosylation of asparagines; positive N-Glyc agreement results (potential >0.5) are reported here, considered high confidence for predictions with 8/9 or 9/9 jury agreement. Viral (NS2B/NS3) and host proteases (furin, signalase, unknown) cleavage enzymes were designated based on previous reviews of proteases associated with orthoflavivirus replication [24–30]. Cysteine bridges and transmembrane domains were predicted based on alignments with other orthoflaviviruses [31–40]. Secondary RNA structures at the flanking 5' and 3' UTR were predicted using the RNA folding form on the UNAFold webserver [41]. Folding parameters included selection of external loop (flat_alt) algorithm [natural (5'UTR), untangle (3'UTR)], and maximum base pair distance of 80 nucleotides for the 3' UTR. Complementary sequence motifs in the 5' and 3' UTR including upstream AUG region (UAR), downstream AUG region (DAR), and cyclization sequence (CS) were predicted based on alignments with published orthoflavivirus sequence elements [42–50].

## Nucleotide sequence accession numbers

The four consensus sequences were uploaded onto NCBI GenBank under the following accession numbers: PV035816 (BeAn 217201), PV035817 (BeAn 4116), PV035818 (BeAn 4073), and PV035818 (CoAr 41922).

## Phylogenetic analysis

The phylogenetic analysis was performed using a curated dataset of eighty-one genomes (prototype strains prioritized)[51–56] retrieved from NCBI GenBank, of mosquito-borne (MB), tick-borne (TB), no known vector (NKV), and a subset of insect specific (IS) orthoflavivirus species, alongside the four novel consensus sequences (S1 Table). MacVector version 18.5.1 was used to isolate the ORF nucleotide sequence, followed by codon translation and amino acid alignment using MAFFT multiple sequence alignment software version 7.271 [57], according to the accuracy-oriented method E-INS-i. Next, a maximum-likelihood (ML) phylogenetic tree was reconstructed in IQ-TREE v. 2.2.2.6 [58] using the best-fit model of amino acid substitution according to the Bayesian information criterion (BIC) inferred by the ModelFinder tool [59]. The reliability of branching patterns was tested using a combination of ultrafast bootstrap (UFBoot) and the SH-like approximate likelihood-ratio test (SH-aLRT) [60,61] with percent support values of 95% and 80%, respectively, considered thresholds of high confidence in nodal support. The final tree was visualized and edited in FigTree 1.4.4.

## Transmission electron microscopy

Individual T-25 flasks were seeded with Vero CCL81 or C6/36 cells and then infected next day with one of three BSQV strains; BeAn 217201, BeAn 4116, or BeAn 4073. Upon observation of cytopathic effect (CPE)(e.g., plaque formation) in Vero CCL81 monolayers (~24 hours), supernatant was removed from both Vero CCL81 and C6/36 cultures, then monolayers immediately fixed and stored at 4 °C for at least 24 hours. Fixation reagent was a mixture of 2.5% formaldehyde prepared from paraformaldehyde powder, and 0.1% glutaraldehyde in 0.05M cacodylate buffer pH 7.3 to which 0.01% picric acid and 0.03% $CaCl_2$ were added. For ultrastructural analysis in ultrathin sections, the monolayers were washed in 0.1 M cacodylate buffer, cells were scraped off and processed further as a pellet. The pellets were post-fixed in 1% $OsO_4$ in 0.1M cacodylate buffer pH 7.3 for 1 hr, washed with distilled water and *en bloc* stained with 2% aqueous uranyl acetate for 20 min at 60°C. The pellets were dehydrated in ethanol, processed through propylene oxide and embedded in Poly/Bed 812 (Polysciences, Warrington, PA). Ultrathin sections were cut on Leica EM UC7 ultramicrotome (Leica Microsystems, Buffalo Grove, IL), stained with lead citrate and examined in a JEM-1400 (JEOL USA, Peabody, MA) transmission electron microscope at 80 kV. Digital images were acquired with a bottom-mounted CCD camera Orius SC200 1 (Gatan, Pleasanton, CA).

## Replication kinetics

Cells were seeded onto 12-well plates with maintenance media (Table 1) to reach ~80% confluency in 1–2 days. BSQV strains (BeAn 217201, BeAn 4116, BeAn 4073) at MOI = 0.01 were used to inoculate cell monolayers by one hour incubation with tilting of plates every 15 min, in the respective 28°C (mosquito cells) or 37°C (vertebrate cells) incubator. Wells were washed three times with PBS and overlaid with 2mL infection media, with 250µL immediately removed and titrated to assess measure carryover virus (e.g., 0 dpi). The entirety of media was collected and replaced with fresh infection media every 24 hours up to 12 dpi. Collected media was clarified by low-speed centrifugation and titrated on Vero CCL81 cells the same day following the described virus quantification protocol. Comparisons of monolayer integrity between BSQV strains were recorded every other day using a CKX53 inverted microscope (Olympus, Waltham, MA) with the 10×CACHN-IPC objective lens (Olympus) and an attached EP50 digital camera (Olympus).

## Statistical analysis

JMP Pro 18 software (JMP Statistical Discovery, Cary, NC, USA) was used for all statistical tests and graphs. Viral titers (PFU/mL) were $\log_{10}$-transformed to assess the overall and daily BSQV strain variation during replication kinetics experiments using two-way repeated measures ANOVA and Tukey's multiple comparisons test. Values below the limit of detection (< 1 plaque produced in Vero monolayer with infection of 100µL undiluted sample) were treated as 0.9 $\log_{10}$ PFU/mL for graphing and statistical purposes. Effects of BSQV strain, day, and the interaction between strain and day were tested.

## Results and discussion

### Genome characterization

NCBI BLAST results confirmed BeAn 217201 and BeAn 4116 strains as BSQV, and ClustalW multiple sequence alignment demonstrated high congruency of both isolates to the archived (NCBI: NC_009026) and novel (PV035818) sequences of the prototype strain, BeAn 4073 (Table 3). The NCBI reference genome differs by seven additional nucleotides in the 3'UTR with the offset appearing between sites 11514–11579. Our four novel consensus sequences have a matched ORF length (3430aa); however, the CoAr 41922 strain has a longer 5'UTR (125nt) and 3'UTR (704nt) compared to the 5'UTR (104nt) and 3'UTR (438nt) of BeAn 217201, BeAn 4116, and BeAn 4073. The CoAr41922 strain shares higher nucleotide (80.93%) and amino acid (95.69%) homology with Naranjal virus (NAJV) (NCBI: KF917538), another member of the *Aroa* virus group (*Orthoflavivirus aroaense*), compared to the other three BSQV strains. The International Committee on Taxonomy of Viruses lists orthoflavivirus species demarcation criteria as (i) nucleotide and deduced amino acid sequence data, (ii) antigenic characteristics, (iii) geographic association, (iv) vector association, (v) host association, (vi) disease association, and (vii) ecological characteristics [62]. With limited ecological knowledge of Aroa serocomplex species, we concluded that CoAr41922 was not BSQV based on genetic homology, and this strain was thereby excluded from further genomic annotation or replication kinetics studies.

### Phylogeny

Each of the four consensus sequences are clustered within the Aroa serocomplex (Fig 1). The BeAn 217201 strain diverged prior to the shared node between BeAn 4073 and BeAn 4116, which displayed the closest evolutionary history. Notably, these two strains were isolated from sequential blood draws, three days apart, from a sentinel non-human primate (*Alouatta belzebul,* red howler monkey) near Belem, Para State, Brazil [1] (Table 1). Meanwhile, BeAn 217201 was isolated from a sylvatic rodent (*Proechimys* sp*.,* spiny rat) within the same region. The CoAr 41922 strain shares a node with NAJV with clear divergence from the BSQV isolates. There are no additional NAJV genomes available, thus the evolutionary history between the NAJV prototype (isolated 1976, Ecuador, sentinel hamster) [27] and CoAr 41922, lacks clarity.

**Table 3. ORF sequence comparisons across Aroa serocomplex.**

| Nucleotide | BeAn 4073 | BeAn 4116 | BeAn 217201 | CoAr 41992 | KF917538 | KF917535 | AY632538 |
|---|---|---|---|---|---|---|---|
| BeAn 4073 | 100.0 | 99.9 | 97.3 | 75.8 | 75.0 | 61.6 | 61.6 |
| BeAn 4116 | 99.9 | 100.0 | 97.3 | 75.8 | 75.0 | 61.7 | 61.7 |
| BeAn 217201 | 97.3 | 97.3 | 100.0 | 76.1 | 75.1 | 61.7 | 61.6 |
| CoAr 41992 | 75.8 | 75.8 | 76.1 | 100.0 | 80.9 | 61.3 | 61.2 |
| KF917538 | 75.0 | 75.0 | 75.1 | 80.9 | 100.0 | 61.8 | 61.3 |
| KF917535 | 61.6 | 61.7 | 61.7 | 61.3 | 61.8 | 100.0 | 76.7 |
| AY632538 | 61.6 | 61.7 | 61.6 | 61.2 | 61.3 | 76.7 | 100.0 |
| **Amino Acid** | **BeAn 4073** | **BeAn 4116** | **BeAn 217201** | **CoAr 41992** | **KF917538** | **KF917535** | **AY632538** |
| BeAn 4073 | 100.0 | 99.9 | 99.4 | 86.9 | 86.8 | 77.5 | 60.8 |
| BeAn 4116 | 99.9 | 100.0 | 99.4 | 86.9 | 86.8 | 77.5 | 60.7 |
| BeAn 217201 | 99.8 | 99.8 | 100.0 | 87.0 | 86.8 | 77.5 | 60.8 |
| CoAr 41992 | 94.9 | 94.9 | 95.0 | 100.0 | 95.7 | 77.3 | 60.6 |
| KF917538 | 94.8 | 94.9 | 94.9 | 98.7 | 100.0 | 77.4 | 76.7 |
| KF917535 | 61.2 | 61.2 | 61.3 | 61.1 | 61.2 | 100.0 | 88.5 |
| AY632538 | 77.0 | 77.0 | 77.0 | 76.6 | 60.9 | 96.0 | 100.0 |

ClustalW multiple sequence alignments were performed with MacVector version 18.5.1 to produce square matrices quantifying both percent sequence identity and sequence similarity of BeAn 217201, BeAn 4116, BeAn 4073, and CoAr 41922 against nucleotide (**top**) or (**bottom**) amino acid ORF sequences against Aroa serocomplex species. Prototype sequences (accession, isolate) retrieved from NCBI GenBank for AROAV (KF917535, Macaray 01809), NAJV (KF917538, 25008), and IGUV (AY632538, SPAn 71686). Similarity scores (%) are shown below the diagonal with identity scores (%) above.

## Transmission electron microscopy

In ultrathin sections of Vero and C6/36 cells, approximately 19 hours post-infection, all examined virus strains displayed similar morphology of intracytoplasmic membranes transformation characteristic of orthoflaviviruses: formation of convoluted membranes (CM) and smooth membrane structures (SMS) within the cisterns of granular endoplasmic reticulum (ER) which represent orthoflavivirus replication complexes (Fig 2A-2C, 2E, and 2F). Within the expanded cisterns of granular ER, immature virions ~40 nm in diameter could be observed (Fig 2C-2E, arrows).

## Genome annotation

**5' and 3'UTRs.** Two strains, BeAn 4073 and BeAn 4116, were 100% identical at the nucleotide level within the 5' and 3' UTR, with six divergent nucleotides compared to BeAn 217201. The characteristic stem-loops, dumbbells, and hairpin structures associated with the folded orthoflavivirus genome [63,64] were visualized for all three strains, with one structure representing both BeAn 4116 and BeAn 4073 (Figs 3 and 4). The BSQV 5' UTR spans nucleotides 1–104; however, for structure prediction software, an additional forty-eight nucleotides following the first AUG codon (up to the second AUG codon) were included. This allowed incorporation of the orthoflavivirus capsid hairpin (CHP) (Fig 3) which is instrumental in first AUG codon selection during viral translation [64]. The previous BSQV 5'UTR structure prediction [65] neglected to include these additional capsid coding nucleotides. There was a single nucleotide change between strains at position 23 that did not impact predicted final structure. The BSQV 5' UTR also includes two conserved stem loop structures; SLA is associated with promoting the viral RNA polymerase and genome capping, while SLB contains the first AUG and the 5' UAR [63,64].

Long-range RNA:RNA interactions, via pairing of short complementary regions (UAR, DAR, CS) between the 5' UTR and 3'UTR, are critical for orthoflavivirus genome circularization and subsequently, orthoflavivirus replication and translation [63,66–68]. The BSQV genome contains these cyclization motifs (S2 Table) in the approximate sites and with similar

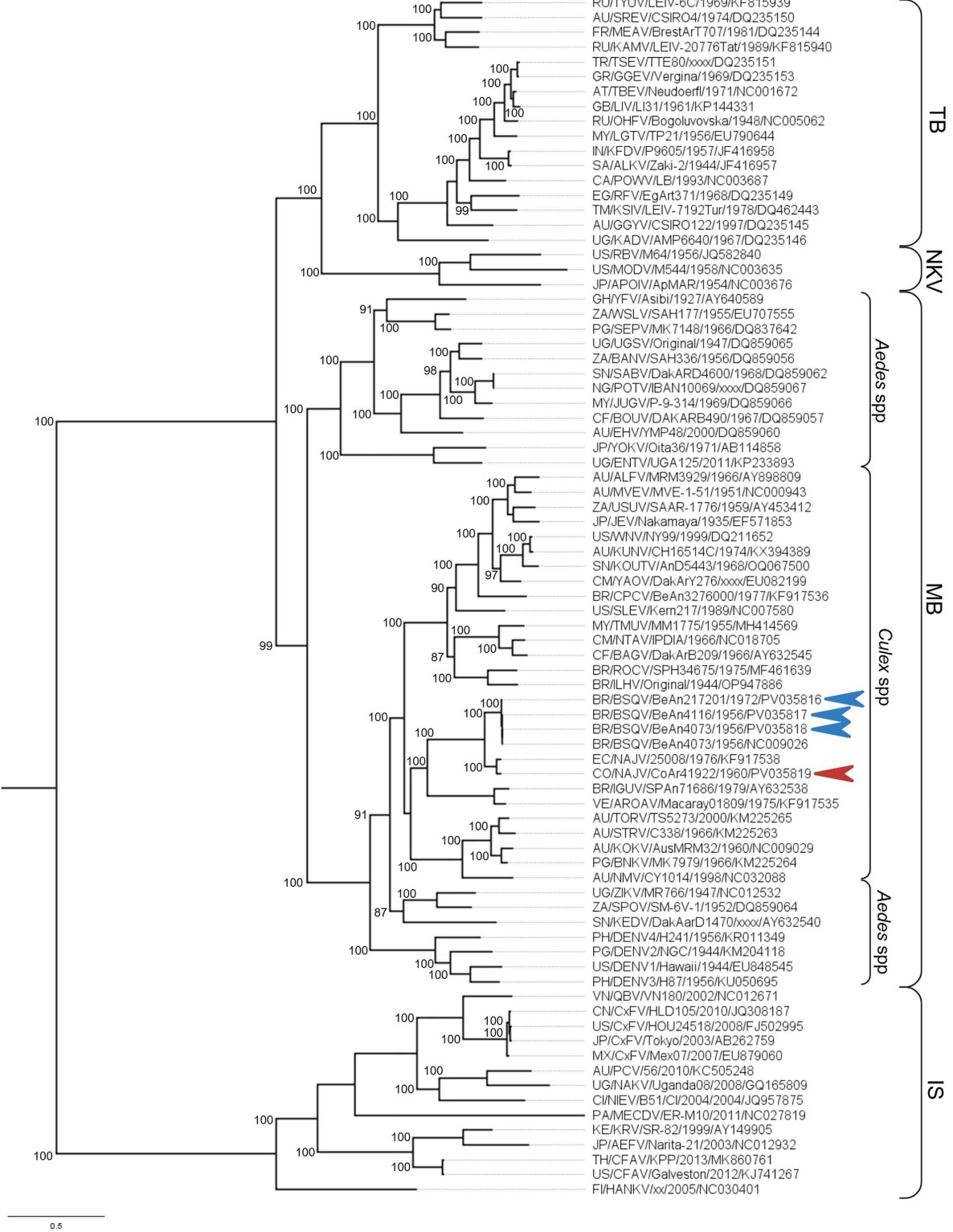

**Fig 1. Phylogenetic relationship of BSQV strains.** Maximum-likelihood tree for the *Orthoflavivirus* genus based on complete amino acid sequences of eighty-one species across four vector-associated clades (right-most brackets); TB, MB, NK, and IS orthoflaviviruses. Numbers in parentheses at major branch nodes are UFBoot (%); only thresholds over 80% are shown. The scale bar depicts a genetic distance of 0.5 or 50% nucleotide sequence

divergence by site. MB subclades of principal vector genera (*Aedes, Culex*) are indicated by innermost brackets. Consensus sequences of confirmed BSQV (n = 3)(blue arrow) and NAJV (n = 1)(red arrow) isolates. Node labels depict country code/ virus name/ isolate/ year/ NCBI accession. Subclades are listed as: mosquito-borne (MB), tick-borne (TB), insect specific (IS), or no known vector (NKV). The final tree was visualized and edited in FigTree v1.4.4, with supplementary brackets, arrows, and labels (node values, vector clade) added at Biorender.com (https://BioRender.com/5ov1u7j).

sequence lengths (UAR: 12–18 nt, DAR: 3–6 nt per domain, CS:10–18 nt) as other orthoflaviviruses [42,50]. Orthoflaviviruses are shown to have one to two DAR [42,69]; BSQV has one DAR domain and the CS and UAR sequence lengths at the range minima.

The BSQV 3' UTR spans nucleotides 10395–10808. Unlike the BSQV 5' UTR, there was variation in predicted RNA folding between strains, as seen with the incongruent stem loop preceding the sHP and terminal 3' SL (Fig 4) due to variation nucleotide positions 10693 and 10694. An orthoflavivirus 3'UTR is composed of three domains (I-III). Domain I is the most variable in architecture and contains tandem SLs that function as exoribonuclease XRN1 resistant RNA (xrRNA) structures [67]. All BSQV strains have one predicted stem loop followed by a shorter hairpin in this domain, indicating one xrRNA (Fig 4) matching previous genus-wide analyses [70,71]. BeAn217201 differs from the other two strains by one nucleotide (10490) in Domain I in the small hairpin (Fig 3).

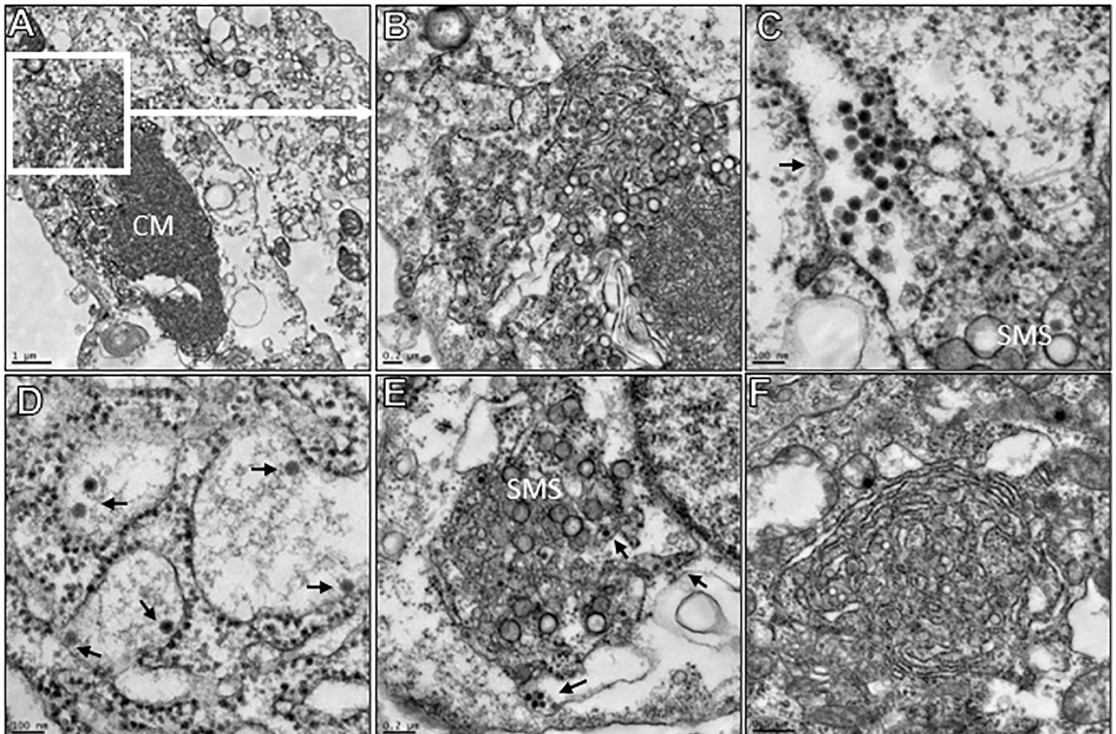

**Fig 2. Ultrastructure of BSQV viruses in ultrathin sections. (A)** BeAn 217201 in Vero cells. Replication complex presented by convoluted membranes (CM) and smooth membrane structures (SMS). Bar = 1 um. **(B)** Fragment of Fig 1A demonstrates immature virions and SMS within the lumen of granular ER. Bar = 0.2 um. **(C)** BeAn 4116 in a Vero cell. Cluster of immature virions (arrow) in a lumen of expanded granular ER. Bar = 100 nm. **(D)** BeAn 4116 in a C6/36 cell. Individual immature virions (arrows) within an expanded cistern of granular ER. Bar = 100 nm. **(E)** BeAn 4073 in a Vero cell. SMS and single immature virions (arrows) within a conglomerate of granular ER cisterns. Bar = 0.2 um. **(F)** BeAn 4073 in a C6/36 cell. SMS within a tight conglomerate of granular ER cisterns. Bar = 0.5 um.

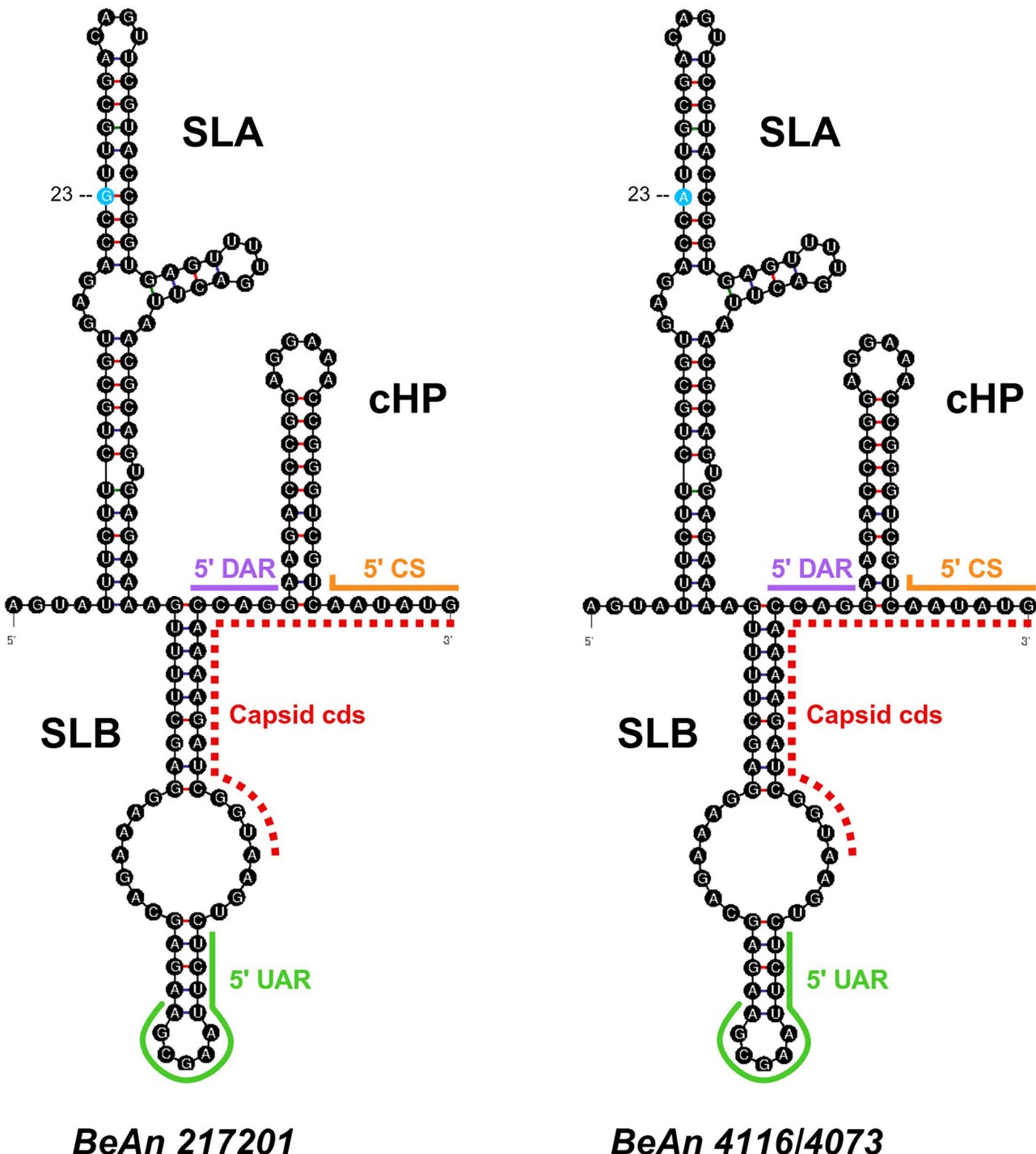

**Fig 3. Predicted 5' UTR secondary structure.** Canonical orthoflavivirus RNA structural elements present in predicted folding of BSQV strains. Conformations include two stem loops (SLA, SLB) and the capsid hairpin (cHP) within the 5' UTR. The capsid protein coding sequence (cds) from the first to second start codon (AUG) was included for software prediction. Nucleotide differences between strains are highlighted in blue. The complementary UAR, DAR, and CS (partial) are color coded to indicate pairing with 3' UTR motifs in Fig 4. The loop free-energy decomposition (ΔG) values for structure predictions are 49.20 (BeAn 217201) and 45.80 (BeAn 4116/4073). Structure prediction originates from UNAFold Web Server and supplementary graphics added at Biorender.com (https://BioRender.com/qtzxxxr).

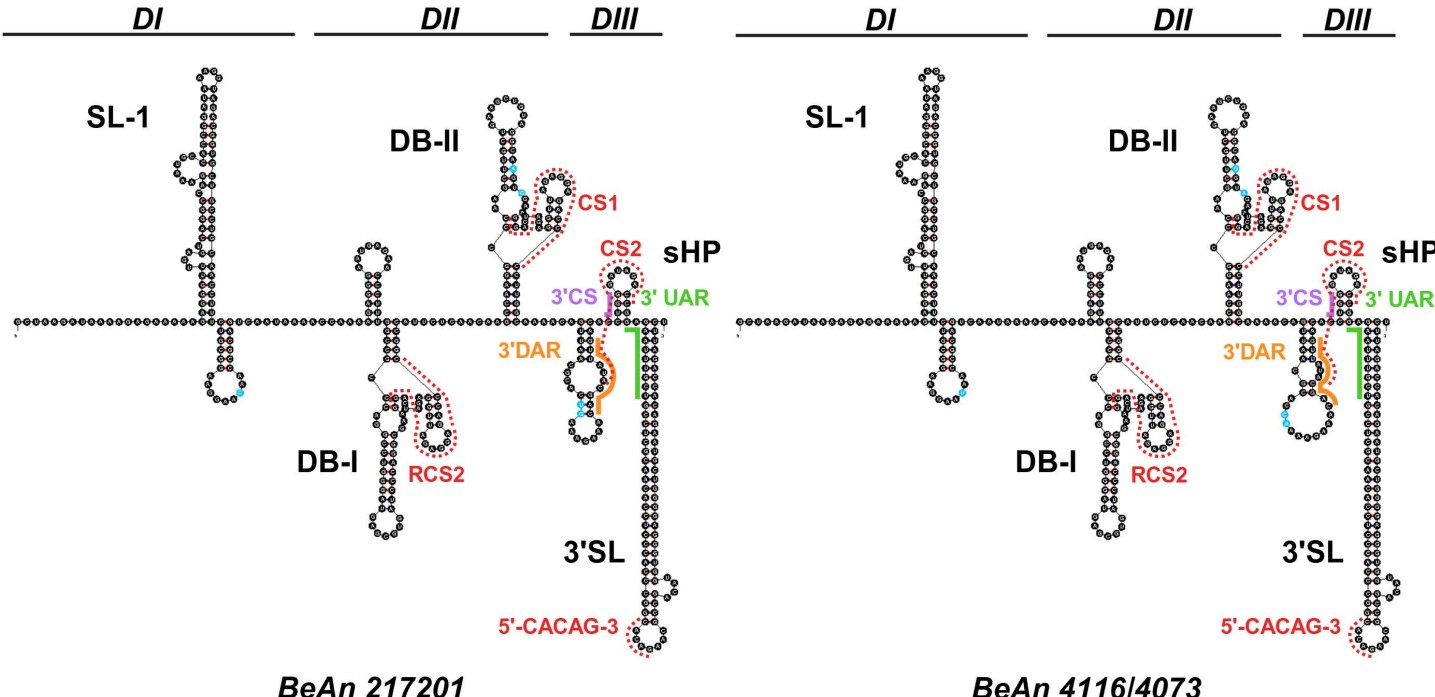

**Fig 4. Predicted 3' UTR secondary structure.** Canonical orthoflavivirus RNA structural elements present in predicted folding include: two stem loops (SL-I, 3'SL), two dumbbells (DB-I, DB-II), and the short hairpin (sHP) across the three domains (DI-III). Nucleotide differences between strains are highlighted in blue. The complementary UAR, DAR, and CS are color coded to indicate pairing with 5' UTR motifs in Fig 3. The conserved 5'CACAG'3' pentanucleotide and RCS/CS sequences are outlined with red dashes. The loop free-energy decomposition (ΔG) values for structure predictions are 136.36 (BeAn 217201) and 132.10 (BeAn 4116/4073). Structure prediction originates from UNAFold Web Server and supplementary graphics added at Biorender.com (https://BioRender.com/2kr46w4).

Domain II also contains structure duplication patterns that vary across orthoflavivirus clades; MB species generally contain 1–2 DBs [67,72]. The BSQV strains differ by two mutations (10641 and 10643) within the DB-II structure (Fig 4). Short direct repeat sequences (CS/RCS) are conserved 3'UTR motifs that cluster primarily within Domain II DBs [73–75] and help maintain efficient rates of RNA synthesis [73], alongside proposed roles in vector-vertebrate transmission [76,77]. Sequence alignment revealed putative BSQV sequences for RCS2 and CS1 on DB I and II, respectively, with CS2 partially straddling the sHP.

Domain III is the most conserved domain. Structural motifs include the short hairpin structure (shP) and the 3'SL with roles in viral replication and viability, alongside interactions with viral and host cell factors [47,66,78,79]. The top of the BSQV 3'SL displays a conserved 5′-CACAG-3′ pentanucleotide motif (Fig 4); a major determinant for orthoflavivirus replication [80–82].

**Open reading frame.** A brief overview of the notable features of the three structural (capsid, pr/M, envelope) and seven nonstructural (NS1, NS2A, NS2B, NS3, NS4A, NS4B, NS5) of BSQV (Fig 5) is described below. BSQV genome annotation across strains (polypeptide boundaries, predicted PTM sites, amino acid differences) is available in S3 Table. Predicted post-translational modifications by gene are listed in Table 4.

**Capsid.** The orthoflavivirus capsid protein is a critical structural component for RNA packaging and virion assembly [84]. The BSQV capsid protein is 126 amino acids long, composed of 107 amino acids at the N-terminus as the core capsid protein, followed by eighteen amino acids residues at the C-terminus functioning as a transmembrane domain (TMD) anchor. There is one sequence variation between BeAn 217201 versus BeAn 4116 and BeAn4073. There are

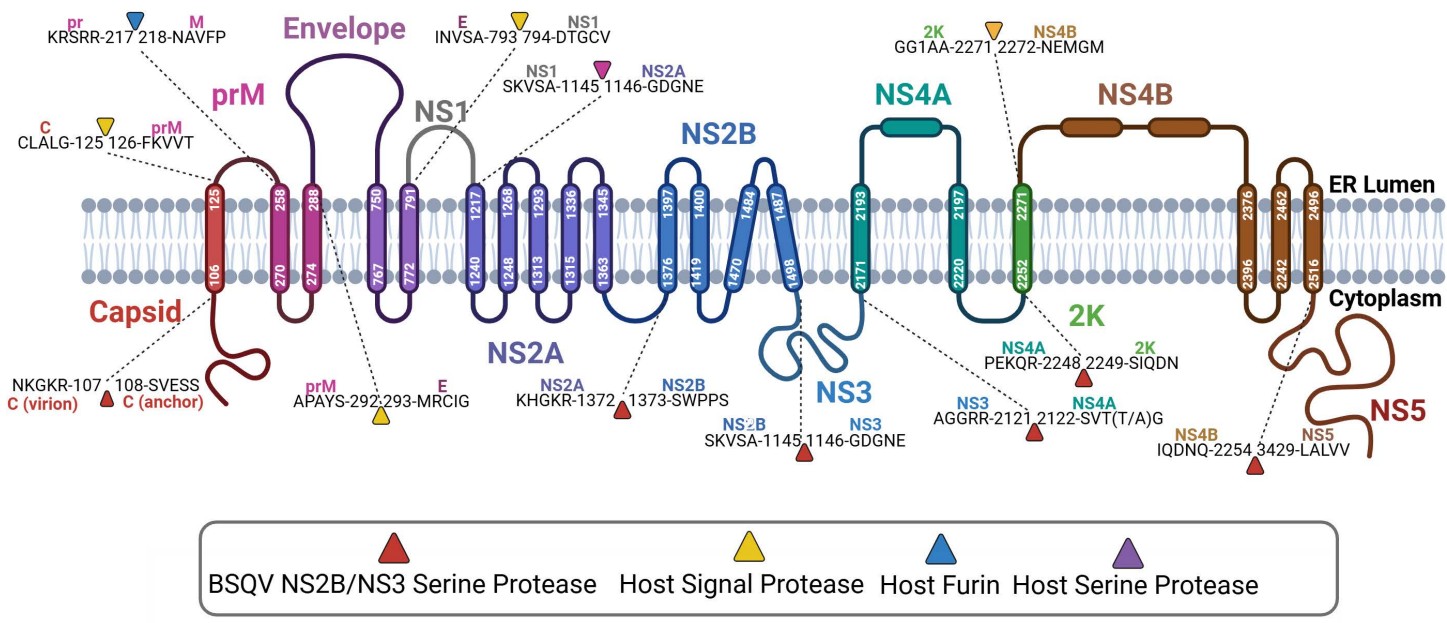

Adapted from *Plante et al 2023*

**Fig 5. BSQV Polyprotein Annotation.** Consensus amino acid sequence across the three BSQV strains. The three nonstructural and seven structural proteins are distinguished by chain color. Polyprotein cleavage sites with the five preceding and five successive residues are defined, with associated proteases distinguished by triangle color. Figure adapted from Plante et al. [83] and created at Biorender.com (https://BioRender.com/jo332fm).

three predicted phosphorylation sites, including the first residue in the TMD and two residues within the cytoplasmic domain.

**prM/M.** Final orthoflavivirus virion maturation occurs in the trans-Golgi network upon furin cleavage of prM [85]. The BSQV prM protein is 157 amino acids long, with a pr peptide of 92 amino acids and an M peptide at the C-terminus with 77 amino acids that spans two predicted TMDs. There is only one sequence variation between BeAn 217201 (threonine) differs from BeAn 4116 and BeAn 4073 (isoleucine). Phosphorylation is predicted at six residues, with only the last site internal to the mature M protein segment. Two N-linked glycosylation sites, one per the pr and M portions, were predicted *in silico* (S3 Table). Six conserved cysteine residues, contributing to structural integrity via intramolecular disulfide bond formation [39,86], aligned to the pr region of BSQV and other orthoflaviviruses in our genus-wide multiple sequence alignment (see data repository).

**Envelope.** The envelope protein composes the external surface of the virion and enables initial viral fusion with the host cell membrane to initiate infection [87,88]. The BSQV envelope protein is 501 amino acids long, with two predicted TMDs, and five variable residues. Phosphorylation was predicted at twenty-one residues. All BSQV strains share the twelve strictly conserved orthoflavivirus cysteine residues [89]. Residue N154 is a common glycosylation site across orthoflaviviruses that contributes to target host cell attachment, virion assembly and secretion, and pathogenesis [90,91]; this site was predicted, albeit with low confidence, only in the BeAn 217201 strain.

**NS1.** The NS1 protein plays multiple roles in genome replication and virion maturation, complement activation and signal transduction, immune stimulation, host immune evasion, and pathogenesis [92–95]. The BSQV NS1 protein is 352

**Table 4. Predicted BSQV Polyprotein Posttranslational Modifications.**

| Gene | Phosphorylation | Glycosylation | Cysteine | Variable Residues |
|---|---|---|---|---|
| Capsid | T101 | | | BeAn 217201 |
| | S91 (BeAn4116, BeAn 4073) | | | 91: S --> N |
| | S108 | | | |
| prM | T15 | N31 | C34 | BeAn 217201 |
| | T82 | N148 | C45 | 86: I --> T |
| | T86 (BeAn 217201) | | C53 | |
| | T90 | | C67 | |
| | S85 | | C69 | |
| | S147 | | C81 | |
| Envelope | T32 | N154 (BeAn 217201) | C295 | BeAn 217201 |
| | T97 (BeAn4116, BeAn 4073) | N498 | C322 | 93: R --> K |
| | T115 (BeAn 4073) | | C352 | BeAn 4116 |
| | T247 | | C366 | 154: N --> S |
| | T310 | | C384 | BeAn 4073 |
| | T320 | | C397 | 120: E --> K |
| | T365 | | C408 | 143: I --> V |
| | S142 | | C413 | 156: S --> L |
| | S153, | | C481 | |
| | S156 (BeAn 4116) | | C582 | |
| | S172 | | C599 | |
| | S174 | | C630" | |
| | S178 | | | |
| | S193 | | | |
| | S209 | | | |
| | S230 | | | |
| | S276 | | | |
| | Y137 | | | |
| | Y182 | | | |
| | Y379 | | | |
| | Y383 | | | |
| NS1 | T2 | N207* | C797 | BeAn 217201 |
| | T87 | | C808 | 175: M --> I |
| | T344 | | C848 | |
| | S45 | | C936 | |
| | S127 | | C972 | |
| | S139 | | C1016 | |
| | S297 | | C1073 | |
| | S300 | | C1084 | |
| | S304 | | C1105 | |
| | S339 | | C1106 | |
| | S351 | | C1109 | |
| | Y32 | | C1122 | |
| | Y113 | | | |

*(Continued)*

**Table 4.** (Continued)

| Gene | Phosphorylation | Glycosylation | Cysteine | Variable Residues |
|------|-----------------|---------------|----------|-------------------|
| NS2A | T50 | N131* | | BeAn 217201 |
| | T100 | N210 | | 202: T -->A |
| | T169 | | | 205: Q -->H |
| | S185 | | | |
| | S197 | | | |
| NS2B | S1 | | | |
| | S5 | | | |
| | S61 | | | |
| | S69 | | | |
| | S72 | | | |
| | S86 | | | |
| | S128 | | | |
| NS3 | T112 | | | BeAn 217201 |
| | T175 | | | 88: R -->H |
| | T184 | | | 109: V -->I |
| | T201 | | | 210: Q -->K |
| | T245 | | | 347: T -->A |
| | T267 | | | |
| | T377 | | | |
| | T393 | | | |
| | S17 | | | |
| | S251 | | | |
| | S253 | | | |
| | S306 | | | |
| | S426 | | | |
| | S547 | | | |
| | S585 | | | |
| | S596 | | | |
| | Y473 | | | |
| NS4A | S1 | | | BeAn 217201 |
| | S127 | | | 4: T -->A |
| | | | | 93: G -->V |
| NS4B | T20 | | | |
| | T250 | | | |
| | S24 | | | |
| | S88 | | | |
| | S193 | | | |

*(Continued)*

**Table 4.** (Continued)

| Gene | Phosphorylation | Glycosylation | Cysteine | Variable Residues |
|---|---|---|---|---|
| NS5 | T93 | N233 | | BeAn 217201 |
| | T161 | | | 197: L -->M |
| | T266 (BeAn 217201) | | | 266: I -->T |
| | T395 | | | 383: R -->K |
| | T421 | | | 563: Q -->K |
| | T543 | | | 644: H -->L |
| | T572 | | | 810: N -->T |
| | T640 | | | 897: g -->E |
| | T751 | | | |
| | T794 | | | |
| | T862 | | | |
| | S56 | | | |
| | S59 | | | |
| | S128 | | | |
| | S214 | | | |
| | S319 | | | |
| | S340 | | | |
| | S342 | | | |
| | S386 | | | |
| | S499, | | | |
| | S523 | | | |
| | S595 | | | |
| | S633 | | | |
| | S665 | | | |
| | S745 | | | |
| | Y22 | | | |
| | Y27 | | | |
| | Y504 | | | |
| | Y883 | | | |

Putative phosphorylation and N-linked glycosylation sites based on *in silico* predictions, and cysteine residues involved in disulfide bond formation, for each gene in the BSQV polypeptide. Variable residue positions are noted, internal to gene, with arrow pointing to variation in specified strain (2nd amino acid) compared to other two strains (1st amino acid). PTM predictions are matched across the three BSQV strains, otherwise indicated by a specific strain within parentheses. * - indicates high confidence for predicted glycosylation site with 8/9 or 9/9 jury agreement.

amino acids long without predicted TMDs. There is one variable residue between BeAn 217201 versus BeAn 4116 and BeAn 4073. A single BSQV NS1 residue had a positive glycosylation prediction, with high confidence, that matched one of two conserved orthoflavivirus glycosylation sites that are critical for NS1 secretion, viral replication, immunogenicity, ER remodeling activity, and neurovirulence [26,92,95–97]; however, only one site had a positive prediction with high confidence. All BSQV strains have thirteen predicted phosphorylation sites, including a match (Y32) to one of two predicted NS1 phosphorylation sites essential for DENV propagation [98]. The twelve conserved cysteine residues [93] aligned between all BSQV strains and other orthoflavivirus species.

**NS2A.** The NS2A protein promotes RNA synthesis, virion assembly, immune evasion, and pathogenesis [36,37,99–104]. The BSQV NS2A protein is 227 amino acids long with two variable residues and five predicted TMDs. BeAn217201 differs from BeAn 4116 and BeAn 4073 at two residues. Phosphorylation was predicted for the same five sites across

all three BSQV strains. Glycosylation was predicted at two residues, one with high confidence by total jury agreement; however, validation or functional analysis of N-linked glycosylation of the NS2A protein has not been reported for other orthoflaviviruses.

**NS2B.** The NS2B is a critical cofactor for the NS3 viral serine protease, alongside roles in viral RNA synthesis, virion assembly, and host immune evasion [28,32,105]. The BSQV NS2B protein is 131 amino acids long with no variable residues and has a predicted 51 amino acid hydrophilic cytoplasmic domain central to its four TMDs. Phosphorylation was predicted for the same seven residues across all three BSQV strains. No N-linked glycosylation sites in the NS2B protein were predicted *in silico*.

**NS3.** The NS3 protein functions as both a serine protease and a NTPase/helicase, with critical roles in RNA replication and RNA strand synthesis [24,106,107]. The BSQV NS3 protein is 618 amino acids, lacks TMDs, and has four variable residues across strains. Phosphorylation was predicted at seventeen residues. No glycosylation sites in the NS3 protein were predicted *in silico*.

**NS4A and 2K.** The NS4A protein is critical for viral replication and cellular membrane remodeling via 2K protein dependent mechanisms [33,108,109]. NS4A also contributes to immune evasion with proposed contributions for persistent orthoflavivirus infections [108,110]. The BSQV NS4A protein is 127 amino acids long and is tailed at the C-terminal by 23 transmembrane residues composing the 2K peptide. Two variable residues between BeAn 217201 versus BeAn 4116 and BeAn 4073 are present in the NS4A region. There are no variable residues in the 2K domain. One phosphorylation site was predicted at each peptide's first amino acid (serine). No glycosylation sites in NS4A or 2K were predicted *in silico*.

**NS4B.** Via an interplay of direct interactions with other nonstructural proteins, the NS4B protein plays roles in membrane remodeling, innate immune evasion, and potentially, coordinating ER-Golgi trafficking of immature virions [35,111,112]. The BSQV NS4B protein is 253 amino acids long with three TMDs and lacking variable residues. Phosphorylation was predicted at the same five sites for each BSQV strain. No N-linked glycosylation sites were positively predicted *in silico.*

**NS5.** The NS5 functions as a methyltransferase responsible for genome capping, as well as a RNA-dependent RNA polymerase with roles in viral replication alongside contributions to the innate immune response [113]. The BSQV NS5 protein is 905 amino acids long with 7 variable residues between BeAn 217201 versus BeAn 4116 and BeAn 4073. Orthoflavivirus NS5 phosphorylation is recognized but minimally understood [114]. Phosphorylation was predicted at twenty-nine residues in BSQV NS5. One N-linked glycosylation site was predicted *in silico*; however, N-linked glycosylation of the NS5 protein is not reported for other orthoflaviviruses.

### Replication kinetics

BSQV replicated robustly in all ten vertebrate cell lines, including the four human cell lines (Fig 6), with peak titers (6.5 - 8.3 $\log_{10}$ PFU/mL) at approximately 3 dpi (S4 Table). BSQV reached similarly high titers in C6/36 and C7-10 (5.6-7.7 $\log_{10}$ PFU/mL) [115,116] cells, which are *Ae. albopictus* cell lines that lack a functional RNAi pathway. Contrastingly, BSQV replicated poorly in Aag2 (*Ae. aegypti*) (peak titer: 1.4 $\log_{10}$ PFU/mL) and U4.4 (*Ae. albopictus*) (peak titer: 2.86 $\log_{10}$ PFU/ mL) which are both RNAi-competent *Aedes* lines [116–118]. This variation in replication is not entirely attributable to RNAi since the virus replicated to high titers in RNAi-competent cells from *Cx. tarsalis* and *Cx. quinquefasciatus* (CT, HSU,CxTr) [118–120]. Overall, productive BSQV replication in the three *Culex* cell lines corroborates the competence bias of *Culex* suggested by field collections [2], while *Aedes* competence remains to be determined, with no reported BSQV isolations from *Aedes* species or confirmatory *in vivo* studies.

Sustained peak viral titer corresponded with cell viability, with peak titer plateaus in susceptible mosquito lines and post-peak declines in vertebrate cells. CPE in mosquito cells, particularly clumping, was attributed to cell overgrowth conditions (S1 File). The lack of mosquito cell ablation correlates to natural arbovirus transmission dynamics that rely on persistent infection and innate immune tolerance in vector species [121–123]. Six vertebrate cell lines had near total (>95%)

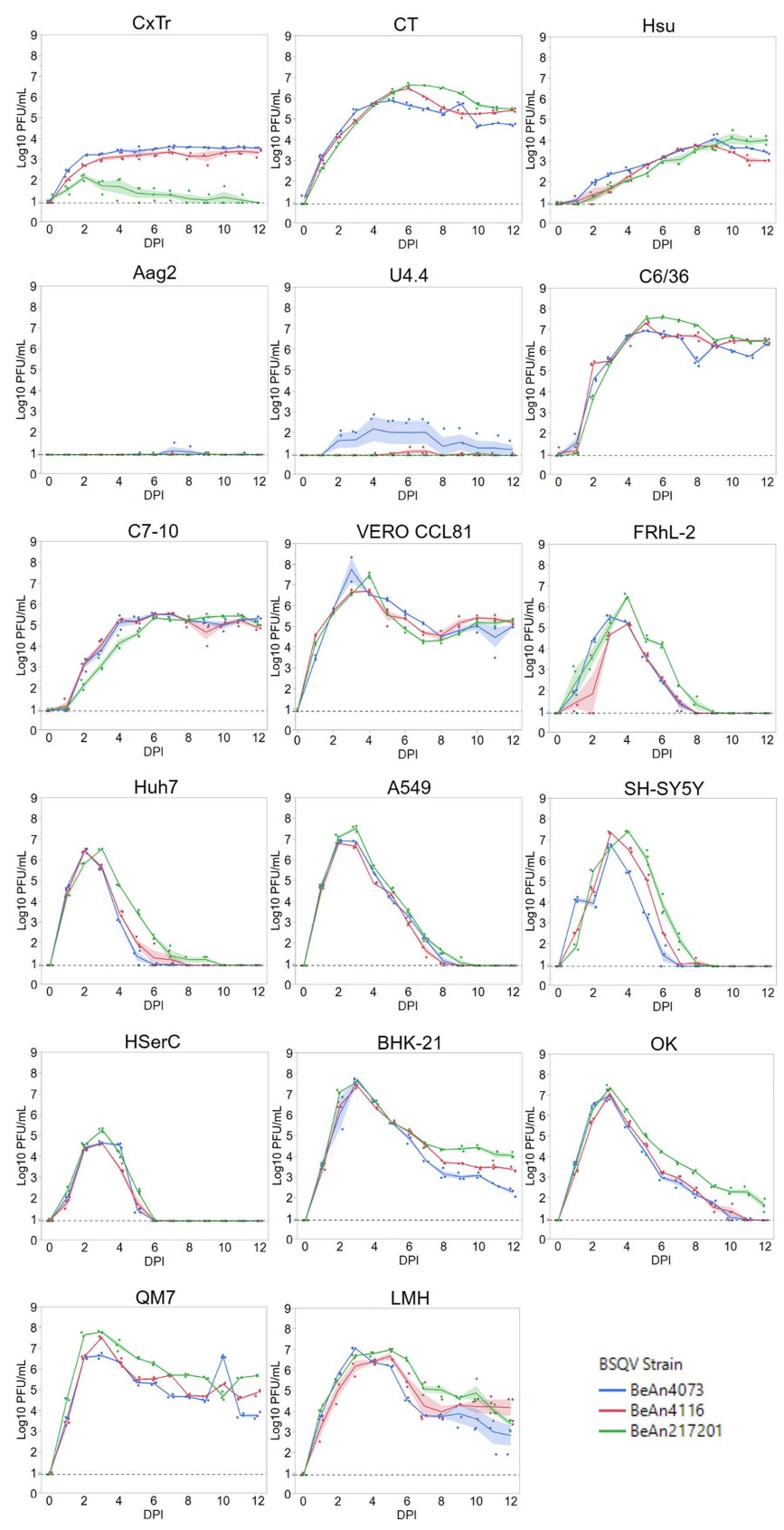

**Fig 6. Replication kinetics.** Kinetic curves of each BSQV strain in mosquito and vertebrate cell lines. Each data point represents the daily mean of three independent replicate wells per BSQV strain per cell line. Dotted lines represent the lower limit of detection (0.09 $\log_{10}$ PFU/mL); data points with no detectable BSQV are represented at the lower limit of detection. BSQV strains are represented by line color, with colored error bands indicating standard error of the mean by dpi.

monolayer ablation: BHK-21, Huh7, OK, Vero CCL81, SH-SY5Y, and LMH, in contrast to four cell lines with either minimal (<5%) (FRhL-2, HSerC) or partial (~50%) (A549, QM7) ablation at endpoint (S1 File). The curtailed CPE in human testicular and lung cells, compared to the extensive destruction of neuroblastoma and liver cells, lends premise for BSQV tissue tropisms, in line with observed liver, brain, and kidney BSQV pathologies in surveillance or laboratory animals [1,5]. SH-SY5Y and HSerC susceptibility need to be considered in context of the blood-brain-barrier and the blood-testis barrier. The capacity of BSQV to cross either of these barriers, as ZIKV does [124,125], in immunologically intact vertebrates remains unknown.

BeAn 217201, BeAn 4116, and BeAn 4073 infection did not produce any observable CPE differences per each cell line. The initial two-way repeated measures ANOVA defined a significant interaction between day post infection and virus strain ($p = 0.00000$-$0.00075$) in all cell lines, indicating significant differences in replication dynamics between strains, except for Aag2 cells ($p = 0.67833$) which had daily titers (0-1.48 $\log_{10}$ PFU/mL) typically below LOD (S4 Table). Post-hoc tests (Tukey connected letter display) delineated specific intra-strain differences in daily viral titer across all cell lines, except Aag2 and U4.4 (S5 Table). Peak BSQV titers were significantly different for each mosquito cell line. For vertebrates, HSerC supported the significantly lowest peak BSQV titers over the 12-day time course. QM7, Vero, SH-SY5Y, OK, A549, and BHK-21 and/or LMH had similar overall peak BSQV titers but demonstrated strain differences with BeAn 217201 having significantly higher peak titers compared to the similar peaks of BeAn 4116 and BeAn 4073, albeit the magnitude of differences in viral titer were small (max difference = 1.6 $\log_{10}$ PFU/ml, 2dpi FRhL-2). Across mosquito and vertebrate cells, BeAn 4073 typically reached peak and/or plateau titers first, whereas BeAn 217201 commonly lagged by one or more days. Future BSQV characterization will benefit from *in vivo* models that investigate these suggested phenotypic differences across strains. Overall, these *in vitro* replication kinetics patterns indicate that BSQV is capable of opportunistic host-range expansion during contact interactions between sylvatic, epizootic, and urban vector-host species.

## Conclusions

We demonstrated that three historical BSQV isolates (BeAn217201, BeAn 4116, BeAn 4073) exhibit the canonical orthoflavivirus characteristics including (i) size and icosahedral morphology, (ii) virion trafficking and host membrane remodeling in mosquito and vertebrate cells, (iii) *Orthoflavivirus* evolutionary relationships, and (iv) genome size and architecture with conserved sequence and structural motifs, and generalized PTM patterns [13,62,126–128]. In addition, BSQV infection was robust across diverse mosquito (peridomestic and urban species) and vertebrate (avian, nonhuman primate, rodent, marsupial, human) cell lines, with notable exceptions potentially related to innate immunity intactness (RNAi, interferon) or mosquito genus (*Culex, Aedes*). Overall, our data support conclusions from historical surveillance that BSQV is a *Culex*-vectored orthoflavivirus with a broad vertebrate and vector range, including species living in proximity to urban communities, suggesting the potential to infect and cause disease in humans.

The limitations of our study include usage of a restricted subset of representative mosquito and vertebrate cell lines to extrapolate a broad vector-host susceptibility. Orthoflavivirus exploitation of PTM machinery is critical to the viral life cycle (142,145–148); our PTM predictions were conducted based primarily on publicly-accessibility of online software tools and thus were not exhaustive. We were only able to investigate the BSQV genome and viral fitness using three available isolates, thereby limiting in-depth genomic meta-analyses of conserved and divergent motifs across strains. Furthermore, BeAn 4073 and BeAn 4116 isolates originated from the same sentinel monkey, via blood collections occurring three days apart [1]. However, BeAn 4073 and BeAn 4116 phenotypic differences cannot be discounted since (i) viral passage

histories differ with additional SM passaging for BeAn 4073 and thus greater potential for murine host adaptation, (ii) consensus sequences from the two viruses have four divergent residues in the Domain I of the envelope protein, and (iii) the two viruses show significant variation in replication kinetics in multiple cell lines. Overall, the precise sequence and structure requirements of predicted sequence, structure, or PTM sites, or the impact of strain differences, towards BSQV replication, translation, pathogenesis, or immune evasion, remain unknown. Lastly, cell line susceptibility does not directly connote transmission potential because: (a) many of the insect cell lines originated from larval stages, and (b) replication kinetics do not reflect host viremia or vector competence outcomes in the context of organismal-level physical barriers or immune factors. Future *in vivo* pathogenesis and transmission studies can enable correlations of BSQV phenotype to genomic annotation data as well as assess capacity of BSQV to sustain urban transmission cycles beyond sporadic spill-over via opportunistic exploitation. Many foundational research studies for neglected orthoflaviviruses coincide with their unprecedented emergence [7,9,11,14,129]. Proactive characterization of arboviruses species like BSQV can be leveraged to facilitate public health response, clinical management, and countermeasure development to combat future emergence scenarios.

## Supporting information

**S1 File. Replication Kinetics - Cytopathic Effects.**
(PDF)

**S1 Table. Orthoflavivirus Genome Sequences Phylogeny.** List of flavivirus sequences used for phylogenetic analysis. Details include accession number, virus abbreviation, strain name, sequence length (nucleotides), location and year of isolation. Subclades are listed as: mosquito-borne (MB), tick-borne (TB), insect specific (IS), or no known vector (NKV).
(XLSX)

**S2 Table. Predicted cyclization and conserved sequence motifs in BSQV UTR.** Predicted RNA nucleotide sequences of (i) complementary UAR, DAR, and CS motifs in the BSQV 5' and 3' genomic termini, and (ii) and conserved orthoflavivirus CS/RCS and penta-nucleotide in the 3' UTR. Lowercase letters indicate a nucleotide with a noncomplementary pairing. *Extension into capsid protein cds.
(XLSX)

**S3 Table. Genome Annotation** - Identification of UTR and polypeptide boundaries and sequences for BeAn217201, BeAn 4116, BeAn 4073, and CoAr41922. Software prediction results for glycosylation and phosphorylation sites in the open reading frames of BeAn217201, BeAn 4116, and BeAn 4073.
(XLSX)

**S4 Table. Replication Kinetics – Titration Data** – Table of individual log titers for each BSQV strain and replicate wells 0–12 dpi in all mosquito (n = 7) and vertebrate (n = 10) cell lines. Table of peak log titer for each replicate per BSQV strain per cell line over kinetics time course.
(XLSX)

**S5 Table. Replication Kinetics – Statistics** - Two-Way RM ANOVA and Tukey-Kramer post-hoc test results of replication kinetics studies analyzing effects on viral titer by day post-infection, strain, and the interaction between day and strain. One-Way ANOVA and Tukey-Kramer post-hoc test results of replication kinetics studies analyzing effects on peak viral titer by cell line and strain, with vertebrate and mosquito cell lines considered as individual groups. Connecting letter display tables color coded to indicate group with non-significant difference in daily or peak viral titers.
(XLSX)

## Acknowledgments

The World Reference Center for Emerging Viruses and Arboviruses provided three of the four BSQV isolates, along-side notations of passage history and cell culture characterization. Jessica A. Plante contributed constructive feedback on RACE techniques. Steven G. Widen and the UTMB Genomics Core performed the initial sequencing workflow and consensus alignment of BSQV. Natalia I. Oliveira da Silva, Marielena V. Saivish, and Taylor A. Strange provided technical assistance during replication kinetics cell culturing and sample processing.

## Author contributions

**Conceptualization:** Mauricio L. Nogueira, Nikos Vasilakis.

**Data curation:** Madeline R. Steck, Cecília A. Banho, Vsevolod L. Popov, Haiping Hao.

**Formal analysis:** Madeline R. Steck, Cecília A. Banho, Vsevolod L. Popov, Kathryn A. Hanley, Nikos Vasilakis.

**Funding acquisition:** Madeline R. Steck, Kathryn A. Hanley, Mauricio L. Nogueira, Nikos Vasilakis.

**Investigation:** Madeline R. Steck, Cecília A. Banho, Vsevolod L. Popov.

**Methodology:** Madeline R. Steck, Cecília A. Banho, Vsevolod L. Popov, Haiping Hao, Kathryn A. Hanley, Nikos Vasilakis.

**Visualization:** Madeline R. Steck.

**Writing – original draft:** Madeline R. Steck, Cecília A. Banho, Vsevolod L. Popov, Haiping Hao, Kathryn A. Hanley, Mauricio L. Nogueira, Nikos Vasilakis.

**Writing – review & editing:** Madeline R. Steck, Cecília A. Banho, Vsevolod L. Popov, Haiping Hao, Kathryn A. Hanley, Mauricio L. Nogueira, Nikos Vasilakis.

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
