## [Decision Letter · Decision Letter 0]

20 Aug 2025

Response to Reviewers
Revised Manuscript with Track Changes
Manuscript

Shaden Kamhawi

co-Editor-in-Chief

Paul Brindley

co-Editor-in-Chief

**Journal Requirements:**

At this stage, the following Authors/Authors require contributions: Madeline Steck. Please ensure that the full contributions of each author are acknowledged in the "Add/Edit/Remove Authors" section of our submission form.

**Reviewers' comments:**

**Key Review Criteria Required for Acceptance?**

**Methods:**

-Are the objectives of the study clearly articulated with a clear testable hypothesis stated?

-Is the study design appropriate to address the stated objectives?

-Is the population clearly described and appropriate for the hypothesis being tested?

-Is the sample size sufficient to ensure adequate power to address the hypothesis being tested?

-Were correct statistical analysis used to support conclusions?

-Are there concerns about ethical or regulatory requirements being met?

Reviewer #1: Yes the authors clearly stated their objectives and executed the studies in a rigorous manner.

Reviewer #2: Methods were well described and suitable for the aims/objectives of this study

Reviewer #3: Study design and methods are appropriate and adequately described.

Reviewer #4: (No Response)

**Results:**

-Does the analysis presented match the analysis plan?

-Are the results clearly and completely presented?

-Are the figures (Tables, Images) of sufficient quality for clarity?

Reviewer #1: The results, figures and tables are clearly presented.

Reviewer #2: Results are clear. Only issue was the long section describing each protein. It would be more apprpriate to focus on what was different about the viruses they characterised.

Reviewer #3: The experimental results are presented and analyzed in an appropriate manner.

Reviewer #4: (No Response)

**Conclusions:**

-Are the conclusions supported by the data presented?

-Are the limitations of analysis clearly described?

-Do the authors discuss how these data can be helpful to advance our understanding of the topic under study?

-Is public health relevance addressed?

Reviewer #1: Yes, although I think the authors could have developed the conclusion/ discussion section more.

Considering their observations that Bussuquara virus didn't replicate well in Aedes cell lines it would have been nice to see some mosquito infections. I know they have the facilities and mosquito colonies to test this.

Reviewer #2: Conclusions are well supported by their findings.

Reviewer #3: The conclusions as currently stated are not adequately supported by the data presented. Limitations are noted but should be more clearly described.

Reviewer #4: (No Response)

**Editorial and Data Presentation Modifications?**

Reviewer #1: (No Response)

Reviewer #2: Minor edits

13. Line 118 replace x with symbol for mutiplication

14. Line 120. Replace u with greek symbol for micro

15. Line 129. Unclear what is meant by next day

16. Line 136 to 137. Sentence can be simplified

17. Lines 159 to 164. Results presented in the methods section. Would suggest moving

18. Line 222. Suggest "confidence in branching pattern" be replaced by confidence in nodal support.

19. Line 244. use capital L for litre (like elsewhere)

20. Figure 1 legend. There are no parentheses in the figure.

21. Line 392 to 393. Please revise sentence

Reviewer #3: The manuscript should be revised to more clearly focus on the novel observations and findings.

Reviewer #4: (No Response)

**Summary and General Comments:**

Reviewer #1: The authors characterized three strains of Bussuquara virus, an orthoflavivirus with potential epidemic potential, using genomic, phylogenetic, microscopic, and cell culture techniques. In general, their results demonstrate it is a stereotypical mosquito-borne orthoflavivirus. Interestingly, they did find that BSQV did not replicate in immune-competent Aedes cell lines while replicating in Culex cell lines just fine. It would have been nice to see some experimental mosquito work to complement their cell culture data. Further, the discussion section/ conclusion could have been further dveloped. Overall, this is a nice descriptive paper.

Reviewer #2: This study has characterised historical BSQV isolates from two hosts from a single location and assessed their potential to infect insect and vertebrate cell lines. I think it is important to note that the authors have made use of approaches that do not require animal studies. Their findings are still relevant and address the aims and outcomes outlined in the manuscript. They confirm three were BSQV-like based on nucleotide identity, virion morphology and cytopathology on several cell lines. Characterisation of their DNA and proteins further adds to the conserved features of orthoflaviviruses and differences in cytopathology between two strains will be valuable information about these strains for future studies. The paper highlights a neglected area of research and would be of interest to those unravelling the importance of insect viruses and vector-borne pathogens of humans and other vertebrates.

Major comments

1. Supplementary tables (including all worksheets) require proper headers that describe their contents.

2. I couldn't access the zenodo link to assess the raw data

3. The manuscript was quite long and the description of each protein-coding region could be significantly summarised to highlight only what was different in the three strains characerised herein.

4. Line 142-144. Was the RNA DNase treated? Would this be required for RT step?

5. Line 154. Provide a list of taxa/protein domains in the custom database in a supplementary table.

6. Line 156 to 158. Add an explanation for the need to align with BWA and bowtie2?

7. Lines 206 to 209 these are results (move down?)

8. The BLAST results in Table 2 report the % identity in the local alignment. It should be clear that some regions of the sequences may not have aligned locally and were not

9. included in this result (when compared to a global alignment reported by BLAT, for example).

10. It appears that BeAn 4073 and 4116 were from the same animal and likely represent the same strain? Would it be more appropriate to say that this study characterised two strains only?

11. Line 363. Here you state that folding differs, but Fig 4 structures look identical apart from some nucleotide differences in blue. Should be clear if differences are with folded templates not shown in the figure.

12. Line 395. You mention post-translational modifications, but you are also showing SNPs that confer a synonymous or non-synonymous change. This should be clear to the reader. Also Table 3 header need to be more descriptive, including that these are all predictions (see methods and discussion).

Reviewer #3: Steck et al report the results of sequencing and in vitro infection using the prototype strain of Bussuquara virus (BSQV, member of Orthoflavivirus aroaense) available through ATCC and the three other strains in the arbovirus reference collection at UTMB. Full genome sequencing was performed and results were annotated and used to determine phylogenetic relationships and to predict RNA secondary structure and protein post-translational modifications. In vitro infection was performed in 7 mosquito cell lines and 10 vertebrate cell lines for measurement of extracellular virus and electron microscopic observations were performed in C6/36 and Vero cells. The main findings were that two of the isolates/strains in the reference collection were confirmed as BSQV whereas the third was identified genetically as Naranjal virus, a related member of Orthoflavivirus aroaense. The three strains of BSQV showed typical genetic characteristics of orthoflaviviruses, induced typical ultrastructural changes in C6/36 and Vero cells, and replicated in most of the mosquito cell lines tested and all of the vertebrate cell lines tested, with minor differences in virus yield between strains but more significant differences in replication between cell lines. The authors main conclusions are that BSQV has genetic characteristics typical of orthoflaviviruses and in vitro replication characteristics that indicate the potential for more extensive epizootic and urban transmission.

The study provides novel information about a poorly characterized arbovirus. The results are generally presented in an appropriate fashion. Given the limited information available on BSQV, the information is likely to be of interest to readers of PLoS Negl Trop Dis. Although the conclusions are largely reasonable, the presentation of the data and conclusions should be improved to better inform the readership.

Specific major comments:

1. Through the manuscript, the authors should conform to the current ICTV nomenclature on BSQV and related members of Orthoflavivirus aroaense. Although the authors acknowledge the current ICTV species designation of the Aroa serocomplex (line 283), the manuscript does not note that all the viruses studied have this same ICTV designation, and NAJV is incorrectly identified as a related species in the Aroa serocomplex (line 267).

2. Key statements in the Title and Abstract are not adequately supported by the research study, specifically that the work provides "clarification of emergence potential" (Title) and that BSQV "has the capacity to enter epizootic and urban transmission cycles" (Abstract). These should be revised to more accurately reflect the work presented.

3. The finding of BSQV replication in most of the cell lines tested appears to be the basis for the authors’ conclusions regarding its emergence potential. However, the study does not present justification for this conclusion. In addition to tempering these conclusions in the Title and Abstract as noted above (see #2), the authors should discuss the evidence for and against this interpretation.

4. The authors should extend the discussion of prior literature on BSQV (genome annotation and in vitro replication) to the other members of Orthoflavivirus aroaense.

5. The manuscript should more clearly focus on the novel findings of this study. For example, given the availability of a full genome sequence of BSQV, description of typical genetic features of orthoflaviviruses can be significantly truncated (lines 320-504).

6. Lines 530-531- Although statistically significant associations were found, the magnitudes of the differences in viral yield between strains were small; this should be noted.

Specific minor comments:

7. The authors should reiterate in the paragraph on limitations (lines 564-566) that two of the strains studied were isolated from same infected host, thus narrowing further the depth of sampling of BSQV.

8. The authors should note the # of cell lines tested in the Abstract.

9. Table 1- It would be helpful to provide the common species names. Tissue culture supplements for CxTr cells should be provided in the table, as is done for the other cell lines.

10. Lines 117-119- The authors should note the titers of the viral stocks.

11. Table 2 (line 123)- It would be helpful to provide the common species names. "SM" should be defined.

12. Line 154- Additional details should be provided on custom dataset(s).

13. Table 2 (line 275; this table is misnumbered)- For the amino acid analysis, the authors should clarify the meaning of values above and below the diagonal (as these are not identical).

14. Figure 2- It is not clear how the authors determined that the virions were immature.

15. Lines 534, 538- What is referenced as "intra-strain" differences (versus "inter-strain")?

Reviewer #4: Steck et al. present a comprehensive characterization of three historical Bussuquara virus isolates (BeAn 217201, BeAn 4116, and BeAn 4073). The study includes a detailed genome characterization, phylogenetic analysis within the Orthoflavivirus genus, and in vitro replication kinetics in a representative set of vertebrate (including four human) and mosquito cell lines. The methods used are well-described and appropriate and the results are clearly presented. The authors combine the results and discussion sections, which works well especially for the genome characterization and direct comparison with other Orthoflavivirus genomes. The paper also contains a clear discussion of the study’s limitations (only three strains from the same geographic region analyzed...). My main critique of this otherwise solid study concerns the title: the claim that the work “clarifies Bussuquara virus emergence potential” feels somewhat overstated and goes beyond what the presented data fully support, particularly in the absence of in vivo data. I recommend either revising the title or more convincingly addressing this claim in the Discussion/Conclusion section of the paper.

PLOS authors have the option to publish the peer review history of their article (what does this mean? ). If published, this will include your full peer review and any attached files.

**Do you want your identity to be public for this peer review?** For information about this choice, including consent withdrawal, please see our Privacy Policy .

Reviewer #1: **Yes: ** Doug Brackney

Reviewer #2: No

Reviewer #3: No

Reviewer #4: No

**Figure resubmission:**

**Reproducibility:** To enhance the reproducibility of your results, we recommend that authors of applicable studies deposit laboratory protocols in protocols.io, where a protocol can be assigned its own identifier (DOI) such that it can be cited independently in the future. Additionally, PLOS ONE offers an option to publish peer-reviewed clinical study protocols. Read more information on sharing protocols at https://plos.org/protocols?utm_medium=editorial-email&utm_source=authorletters&utm_campaign=protocols

---

## [Decision Letter · Decision Letter 1]

19 Nov 2025

Dear Prof. Vasilakis,

We are pleased to inform you that your manuscript 'Hiding in plain sight: genomic and phenotypic characterization of mosquito-borne Bussuquara virus' has been provisionally accepted for publication in PLOS Neglected Tropical Diseases.

Best regards,

Luis Adrián Diaz, Ph.D.

Academic Editor

David Safronetz

Section Editor

Shaden Kamhawi

co-Editor-in-Chief

Paul Brindley

co-Editor-in-Chief

Dear Authors,

your R1 version of your manuscript has been returned as Accepted from 4 reviewers.

Please considere the following observation of one of the reviewers:

"I would request that the authors re-check entries in the section of Table 2 on amino acid analysis because some pairwise comparisons show a higher percentage sequence identity than percentage sequence similarity (e.g., BeAn 4073 vs. KF917535)."

Reviewer's Responses to Questions

**Key Review Criteria Required for Acceptance?**

**Methods**

-Are the objectives of the study clearly articulated with a clear testable hypothesis stated?

-Is the study design appropriate to address the stated objectives?

-Is the population clearly described and appropriate for the hypothesis being tested?

-Is the sample size sufficient to ensure adequate power to address the hypothesis being tested?

-Were correct statistical analysis used to support conclusions?

-Are there concerns about ethical or regulatory requirements being met?

Reviewer #2: R1 methods are improved and acceptable

Reviewer #3: (No Response)

**Results**

-Does the analysis presented match the analysis plan?

-Are the results clearly and completely presented?

-Are the figures (Tables, Images) of sufficient quality for clarity?

Reviewer #2: R1 results are now clear and acceptable

Reviewer #3: (No Response)

**Conclusions**

-Are the conclusions supported by the data presented?

-Are the limitations of analysis clearly described?

-Do the authors discuss how these data can be helpful to advance our understanding of the topic under study?

-Is public health relevance addressed?

Reviewer #2: (No Response)

Reviewer #3: (No Response)

**Editorial and Data Presentation Modifications?**

Reviewer #2: No further changes suggested.

Reviewer #3: I would request that the authors re-check entries in the section of Table 2 on amino acid analysis because some pairwise comparisons show a higher percentage sequence identity than percentage sequence similarity (e.g., BeAn 4073 vs. KF917535).

**Summary and General Comments**

Reviewer #2: Thank you for your comprehensive response to comments raised in R0. I think the authors have done a good job addressing my comments.

Reviewer #3: The revised manuscript adequately addresses the major comments and (most of) the minor comments raised in the prior review.

PLOS authors have the option to publish the peer review history of their article (what does this mean? ). If published, this will include your full peer review and any attached files.

**Do you want your identity to be public for this peer review?** For information about this choice, including consent withdrawal, please see our Privacy Policy .

Reviewer #2: No

Reviewer #3: No

---

## [Editor Report · Acceptance letter]

Dear Prof. Vasilakis,

We are delighted to inform you that your manuscript, "Hiding in plain sight: genomic and phenotypic characterization of mosquito-borne Bussuquara virus," has been formally accepted for publication in PLOS Neglected Tropical Diseases.

Best regards,

Shaden Kamhawi

co-Editor-in-Chief

Paul Brindley

co-Editor-in-Chief
